# Transition Probability Test for an RO-Based Generator and the Relevance between the Randomness and the Number of ROs

**DOI:** 10.3390/e24060780

**Published:** 2022-05-31

**Authors:** Yuta Kodera, Ryoichi Sato, Md. Arshad Ali, Takuya Kusaka, Yasuyuki Nogami

**Affiliations:** 1Graduate School of Natural Science and Technology, Okayama University, Okayama 700-8530, Japan; ryoichi_satoh@s.okayama-u.ac.jp (R.S.); kusaka-t@okayama-u.ac.jp (T.K.); yasuyuki.nogami@okayama-u.ac.jp (Y.N.); 2Department of Computer Science and Engineering, Hajee Mohammad Danesh Science and Technology University (HSTU), Dinajpur 5200, Bangladesh; arshad@hstu.ac.bd

**Keywords:** true random number generator, ring oscillator, Markov process, hypothesis testing

## Abstract

A ring oscillator is a well-known circuit used for generating random numbers, and interested readers can find many research results concerning the evaluation of the randomness with a packaged test suit. However, the authors think there is room for evaluating the unpredictability of a sequence from another viewpoint. In this paper, the authors focus on Wold’s RO-based generator and propose a statistical test to numerically evaluate the randomness of the RO-based generator. The test adopts the state transition probabilities in a Markov process and is designed to check the uniformity of the probabilities based on hypothesis testing. As a result, it is found that the RO-based generator yields a biased output from the viewpoint of the transition probability if the number of ROs is small. More precisely, the transitions 01→01 and 11→11 happen frequently when the number *l* of ROs is less than or equal to 10. In this sense, l>10 is recommended for use in any application, though a packaged test suit is passed. Thus, the authors believe that the proposed test contributes to evaluating the unpredictability of a sequence when used together with available statistical test suits, such as NIST SP800-22.

## 1. Introduction

The study of finding entropy sources is a traditional and essential topic, with attractive randomness in some applications such as key generation and issuing identifiers in the cryptographic field, for example. In practice, the physical inputs or characteristics of an I/O device on a computer, such as a keyboard or a computer mouse, are well-established sources. However, such inputs are not always ideal; for example, a human-related source such as the input from a keyboard would be affected by the user’s intention. Since the entropy source should be truly random, researchers have investigated and developed other methods using physical phenomena to overcome these drawbacks.

Researchers and developers have paid much attention to making random number generators (RNG) using digital circuits compact, so that they can be implemented together with other modules. There are two main types of circuits that can easily cause unstable signals. One of them is called metastability [1], which is an intermediate state between high and low and is dealt with as a malfunction of a circuit in product development. However, it is known to have the ideal characteristics to act as an entropy source, and can be implemented with just a pair of NAND gates on a field programmable gate array (FPGA), for example. However, since some time is consumed to converge the vibration during metastability, a generator using metastability sometimes faces problems with efficiency.

The other circuit is an oscillation circuit, called a ring oscillator (RO), consisting of NOT gates that are aligned in a ring shape. Sunar et al. introduced an RNG using the ROs in [2]. It was designed to mix the output of multiple RO circuits by using the XOR operation, the output of which is synchronized by an internal clock. Though the construction allows bits to be sampled faster than by using metastability, its randomness was required to be discussed further, since the randomness is easily affected by the number of NOT gates and RO circuits.

Wold et al. [3] extended Sunar’s proposed RNG circuit in which the respective output of ROs is synchronized by delay flip flops (D-FFs). Since the D-FFs contribute to improving the randomness of RO-based generators, Wold’s construction is now widely adopted as an entropy source in various situations. Research on these RO-based generators has been approached from several viewpoints, such as randomness, security, and energy efficiency. The readers can refer to [4,5,6,7,8,9,10,11,12] for further results about extensions of the RO-based generators and randomness evaluations, for example.

In this paper, the authors focus on Wold’s construction and discuss the distribution property of the generator. Furthermore, the relevance between the number of ROs and the quality of randomness is also considered. More precisely, the target concerning the distribution property is the transition probability of bits introduced in the Markov process. This differs from the elements of several famous statistical tests in the sense that the authors’ proposed method discusses the uniformity of a sequence from the relevance of bits at time *t* and t+1, for instance. The authors think this approach contributes to a different aspect of a sequence, together with the currently proposed statistical test suites. In addition, in this paper the authors conduct the same test for generators set up with different numbers of ROs. As a result, it is found that there is a significant relationship between the number of ROs and the randomness property.

This paper is organized as follows: Section 2 introduces the fundamentals related to this work, for example, details of the RO-based generator and statistical tests. Section 3 and Section 4 propose a test for an RNG designed using ROs, and give experimental observations for some generators with different numbers of ROs, respectively. Finally, Section 5 concludes this paper.

## 2. Preliminaries

This paper focused on the probability test for an RO-based generator, as well as the relationship between the randomness property and the number of ROs. This section briefly reviews the idea of a true random number generator based on a ring oscillator, Markov process, and hypothesis testing.

### 2.1. True Random Number Generator Based on Ring Oscillator

This section briefly reviews the fundamentals of an RNG based on ring oscillators, and related works. An RNG is simply referred to as a generator in this paper.

#### 2.1.1. Random Number Generator and Ring Oscillators

RNGs are typically classified into two main classes [13]. One of them is the deterministic RNGs, called pseudorandom number generators (PRNGs). They work algorithmically with a given seed value. Another class is the non-deterministic RNGs, which often adopt some non-reproducible phenomena to generate an ideal random number sequence. Such an ideal sequence is called the true random number generator (TRNG), and many approaches using physical phenomena, called physical RNGs in what follows, have been proposed as a class of TRNG. However, not every RNG can be dealt with as a TRNG, even if it employs a physical phenomena. In this paper, the authors mainly work on evaluating a well-known physical RNG construction from the viewpoint of the unpredictability of sequences.

A representative construction is to use digital circuits to obtain a sequence of digits including bits. There are several approaches such as using noises or the unstable behavior of logic gates. Among them, an oscillation circuit consisting of odd numbers of NOT gates, called a ring oscillator (RO), is widely adopted and studied. As shown in Figure 1, an RO is a circuit that is composed of odd numbers of NOT gates connected in a ring shape.

Since it can be implemented as a logic circuit in an FPGA, a generator based on ROs is cheaper than one that uses external equipment to observe phenomena to obtain a sequence.

Though external equipment may possibly be interfered with or intermediated by an attacker and leak a sequence as a result, a circuit closed inside an FPGA has an advantage in this regard. In detail, without an adequate environment, data through a circuit are hard to eavesdrop directly. In addition, a generator consisting of a logic circuit is preferred in practical use since it can be embedded with other circuits into a chip.

The output of an RO oscillates due to the recursive input from the right edge NOT gate, as the name stands for. The frequency $f_{t_{\mathrm{NOT}}}$ is known to be given by Equation (Equation 1), where $t_{pd}$ and $t_{\mathrm{NOT}}$ denote the propagation delay time of a NOT gate and the number of NOT gates, respectively.
(1)$$\begin{array}{c} f_{t_{\mathrm{NOT}}}=\frac{1}{2t_{pd}t_{\mathrm{NOT}}} \end{array}$$

It is noted that since the propagation delay time changes from high, say $t_{pLH}$, and high to low, say $t_{pHL}$, are the same if CMOS devices are used, we can assume $t_{pLH}=t_{pHL}$ and denote the propagation delay time by $t_{pd}$ for simplicity.

Since an RO is not stable because of effects from the external environment, such as thermal noises, for example, the actual oscillation period has time difference $T_{d}$ from the theoretical oscillation period *T*, as shown in Figure 2, where $T_{d}\ll T$.

Thus, the oscillation period $T_{RO}$ of the RO is given by $T_{RO}=T\pm T_{d}$.

This instability is useful for sampling binary symbols, e.g., 0 and 1, and the circuit size can be scalable depending on the number of NOT gates. Therefore, many researchers have focused on ROs to develop a compact and ideal RNG, based on ROs.

#### 2.1.2. Related Works

Sunar et al. introduced ROs to construct a TRNG in [2]. It was designed to mix the output of the respective RO circuit by using the XOR operation, the output of which is synchronized by an internal clock, as shown in Figure 3.

In [2], Sunar et al. also discussed several properties, such as the ideal length of a ring, from the theoretical viewpoint.

Based on their construction, Wold et al. [3] extended the circuit to be as shown in Figure 4. Wold et al. successfully enhanced the randomness by inserting D-FFs between ROs and an XOR gate to sample the wave endowed by the ROs. They found that short ROs are better for improving randomness, since the difference in the wave frequency can be easily induced by the restriction of the length. In this paper, the authors mainly deal with Wold’s construction to investigate the randomness of continuous digits. In addition, the readers can refer to the results in [4,5,6,7,8,9,10,11,12] concerning RO-based generators and evaluations for more information.

In a previous work [14], the authors discussed the importance of an XOR gate in a generator based on ROs by approximating the periodicity and investigating the transition probabilities of 2-bit patterns. It was revealed that an XOR gate in an RO-based generator contributes to extending the period length, and the number of ROs is of relevance to the distribution property. In this paper, the authors extend this discussion to the statistical randomness evaluation by using the Markov process and hypothesis testing.

### 2.2. Markov Process

A Markov process is a stochastic extension of a finite automaton for which state transitions happen probabilistically. It has a memoryless property, which is to say that any additional information about the future behavior of the process cannot be obtained from the past processes in a random process. More precisely, $X_{1},X_{2},X_{3},\ldots$ are random variables and $P\left\{X|Y\right\}$ denotes the conditional probability of *X* given *Y*. Let $S=\left\{s_{1},s_{2},s_{3},\ldots \right\}$ be the state space and let $p_{i,j}=P\left\{X_{n}=s_{j}|X_{n-1}=s_{i}\right\}$ be the transit probability from $s_{i}$ to $s_{j}$ for a positive integer n>1. The memoryless property, referred to as the Markov property, holds the equality as follows:(2)$$\begin{array}{c} P\left(X_{n}=x_{n}|X_{n-1}=x_{n-1},X_{n-2}=x_{n-2},\ldots ,X_{1}=x_{1}\right)=P\left(X_{n}=x_{n}|X_{n-1}=x_{n-1}\right), \end{array}$$
where $x_{i}\in S$. Such a random process utilizing the Markov property is called a Markov process.

Based on the definition of the Markov process, a Markov chain is defined as a Markov process with discrete time and discrete state space. Thus, a Markov chain is a discrete sequence of states, denoted by *S*, with random variables $X_{1},X_{2},X_{3},\ldots$ such that the probability of any given state $X_{n}$ only depends on the current state $X_{n-1}$, as shown in Equation (Equation 2). The process diagram of the Markov chain is a directed graph describing the Markov process. For example, a simple two-state Markov chain can be illustrated as shown in Figure 5.

### 2.3. Hypothesis Testing and Z-Test

Hypothesis testing is a method of testing whether claims or hypotheses concerning a population are likely to be true. There are two hypotheses: the null hypothesis and an alternative hypothesis. The null hypothesis is a statement about a population parameter, which is assumed to be true. In contradiction to the null hypothesis, the alternative hypothesis is a statement that says the value of the population parameter does not match the value in the null hypothesis.

Hypothesis testing is conducted by following the steps summarized below.

State a null hypothesis and alternative hypothesis;Select a random sample from the population;Set a significance level and perform an appropriate statistical test;Decide whether the null hypothesis is valid or not.

The significance level is a criterion to decide the value stated in the null hypothesis. The decision often comes from the outcome of the statistical test, using the *p*-value, which is the probability of obtaining a sample result under the null hypothesis, which is then compared to the significance level.

A Z-test is a hypothesis test in which the Z-score, also called the Z-statistic, follows a normal distribution. It determines whether the mean of random variables $X_{1},X_{2},\ldots ,X_{n}$ is equal to a mean $m_{0}$ when the variances of $X_{i}\left(1\leq i\leq n\right)$ are known. It is noted that the test is considered to be accurate if $X_{k}$ follows the normal distribution, or to be approximately accurate if *n* is sufficiently large (for example, n≥30). The test is conducted by assuming that the Z-score follows the standard normal distribution. It is calculated by
$$\begin{array}{c} Z=\frac{\bar{X}-m_{0}}{\frac{\sigma }{\sqrt{n}}}, \end{array}$$
where $\bar{X}=\sum_{i=1}^{n}X_{i}/n$ and σ denote the standard deviation.

## 3. A Test Method and Evaluation Process

This section introduces the details of the test and its evaluation process.

### 3.1. Background of the Proposed Test

Typically, a sequence generated by PRNGs or TRNGs is evaluated by statistical tests such as TestU01 [15] and NIST SP 800-22 [16]. These are packages of several statistical tests, and users can smoothly check the statistical randomness by running them. For example, the distribution property of a sequence is evaluated by counting the number of bits or comparing a bit pattern with a template. These evaluations are an inseparable part of the distribution property. However, the authors feel that these evaluations cannot fully cover the features of the property.

In this context, the authors introduce the transition probability by considering the Markov process for an RO-based generator from previous research [14]. It is noted that the readers can refer to [4], as a work related to this paper. In brief, this paper focuses on the 2-bit patterns and deals with them in the state $S=\left\{00,01,10,11\right\}$, with transition probabilities between each other, where each pattern is derived by splitting a sequence of bits from the beginning without any duplication of the index. Furthermore, the authors extend the discussion of this approach to investigate the properties of an RO-based generator in the following sections.

### 3.2. Design of a Test

This section briefly introduces the assumptions and process of the test, including evaluation.

#### 3.2.1. Assumptions

In this paper, a 100 MHz clock is used to sample a bit to generate a sequence with an RO-based generator. The state considered in the Markov process is a 2-bit pattern. Therefore, the time space is a discrete set $T=\left\{2\times 10^{-8},4\times 10^{-8},6\times 10^{-8},\ldots \right\}$.

An RO-based generator is implemented on an FPGA as a combination of lookup tables (LUT) and D-FFs. Hence, both NOT and XOR gates can be expressed by LUTs. Since the wire length causes a difference in RO circuits, this paper intentionally arranges each element so that they can be in the same condition.

#### 3.2.2. Process of the Test and Evaluation

The test conducted in the next section is composed of three steps, as follows:Set the null hypothesis such that the transition probabilities are equal to 1/4;Repeat the following sampling and preparation step 1000 times:(a)Generate a sequence of length 1Kbits on an FPGA, and repeat it 1000 times to obtain a sample sequence of length 1Mbits in total;(b)Split the sequence into 2 bits and calculate the transition probabilities;(c)Observe the distribution of probabilities (it should follow the normal distribution) and decide the significance level;(d)Conduct the Z-test and calculate *p*-values.Discussion

First, the assumption of the null hypothesis is clear from the fact that the ideal distribution of 2 bits is the uniform distribution, having an apparent probability of $1/2^{2}$. Since one of the motivations in this work is to investigate the uniformity of transition probabilities from every pattern, the authors propose sampling a sequence of length 1K bits 1000 times to obtain a *p*-value. By repeating the collection of *p*-values 1000 times, the authors decide whether the null hypothesis is approved or not.

In addition, the authors’ other motivation is to reveal the relationship between the randomness of sequences and the number of ROs in a generator. Several experimental results for different numbers of ROs are introduced in the next section. It is noted that the authors use the Z-test in the following experiments since the number of samples is large (1000 samples). However, a *t*-test should be utilized when the readers need more strict and practical evaluation.

## 4. Experimental Results and Considerations

In this section, the authors show the experimental results of the proposed method. For simplicity, the null hypothesis $H_{0}$ and alternative hypothesis $H_{1}$ throughout the experiment are $H_{0}:\mu =0.25$ and $H_{1}:\mu \neq 0.25$, respectively, where μ denotes the mean of transition probabilities. The FPGA board used to implement the RO-based generator was a Nexys A7-100T Artix-7 series [17], with every RO circuit being composed of only three NOT gates.

### 4.1. Observation

First, let us begin by briefly confirming whether the probability distribution follows the normal distribution. Figure 6, Figure 7 and Figure 8 shows the histograms of transition probabilities when the numbers of ROs are 2, 10, and 20.

By observing the figures, it is apparent that the probability distribution follows the normal distribution as the expectation gradually closes to 0.25, depending on the increment in the number of ROs. Thus, the proposed method can be considered applicable to an RO-based generator.

### 4.2. Comparison of p-Values and Considerations

Based on the previous observation, the authors conducted a Z-test and compared the *p*-values obtained throughout the experiment. It is noted that the significance level is set to 1% to validate the hypothesis more strictly.

Table 1 shows the comparisons of *p*-values obtained by testing sequences generated with *l* number of ROs, where 1≤l≤25. The element highlighted in red denotes the cases in which the *p*-value is less than 0.005, and the blue ones the elements greater than or equal to 0.005, respectively. As seen from the table, the null hypothesis tended to be accepted when l>10. Additionally, compared to the other transitions, the null hypotheses for the specific transitions, such as 01 to 01 and 11 to 11 for l>10, are found to be rejected frequently. To check these assumptions further, the authors conducted additional experiments, as given in the next section.

### 4.3. Post Evaluations for Consideration

The following two characteristics are found throughout the above comparison; thus, the authors carried out further experiments to confirm the likelihood.

-The null hypothesis $H_{0}:\mu =0.25$ is accepted when l>10;-The null hypothesis concerning the transition probabilities from 01 to 01 and from 11 to 11 are often rejected.

The first assumption shows that the mean of transition probabilities is 0.25 when l>10, which is assumed to be an ideal result for the authors. On the other hand, the second points out that the specific transitions do not happen uniformly.

The authors conduct additional experiments from different viewpoints, as described below, to confirm these assumptions.

Comparing the number of sequences that could pass the test (1000 trials);Comparing the difference in the results when the number of NOT gates in an OR circuit is changed (1000 trials).

The experimental results in terms of transition probabilities for each state are introduced in Figure 9, Figure 10, Figure 11 and Figure 12. The horizontal and vertical axes reflect the number of ROs and the number of sequences of length 1 Mbits for which transition probabilities were able to pass the hypothesis test. As seen from the graphs, the number of sequences will gradually become flat for l>10. However, Figure 10 and Figure 12 also show that the transitions from 01 to 01 and from 11 to 11 are relatively low compared with the other transitions.

In the same way, experiments were conducted with the RO-based generator consisting of seven NOT gates. The results are shown in Figure 13, Figure 14, Figure 15 and Figure 16. Comparing the figures in Figure 9, Figure 10, Figure 11, Figure 12, Figure 13, Figure 14, Figure 15 and Figure 16, the readers can find similar characteristics in both graphs, and the assumptions mentioned above are considered to be true.

## 5. Conclusions

This paper focused on an RO-based generator originally proposed by Wold et al.; in addition, the authors proposed a statistical test regarding the state transition probabilities of 2 bits for the generator. The purpose of such a test is to check the uniformity of the respective transition patterns, such as 00,01,10, and 11. This statistical test was applied for RO-based generators consisting of different numbers of ROs. As a result, it was found that the randomness of an RO-based generator depends on the number *l* of ROs, and the result shows that *l* should be larger than 10.

Therefore, the authors successfully evaluated the randomness of RO-based generators numerically, and we can conclude from this study that the circuit of RO-based generators becomes complex depending on the increment in ROs. In addition, it tells us that the users have to recognize the bias hidden in the transition probabilities, especially for practical use.

However, since this paper only focused on the 2-bit case and did not formulate the experimental results, the authors would like to utilize larger bit patterns and different boards to explore the characteristic equation in future works. 

## Figures and Tables

**Figure 1 entropy-24-00780-f001:**
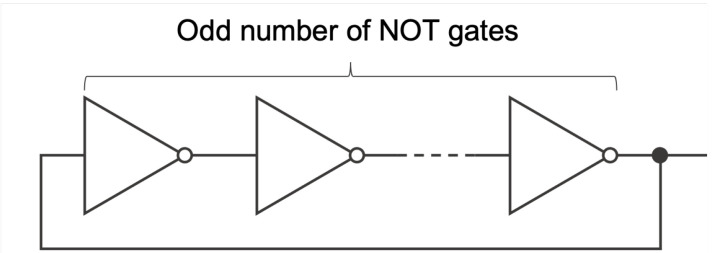
Illustration of a ring oscillator.

**Figure 2 entropy-24-00780-f002:**
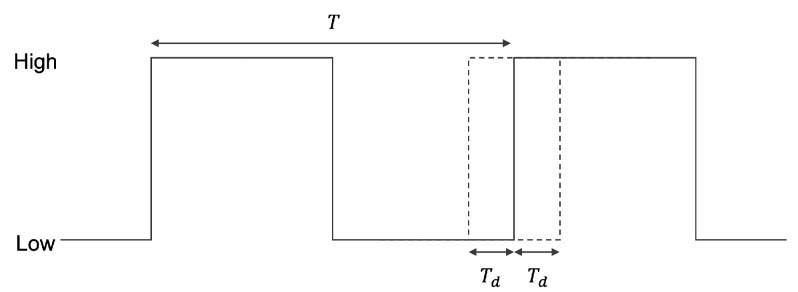
Illustration of an oscillation period in a ring oscillator.

**Figure 3 entropy-24-00780-f003:**
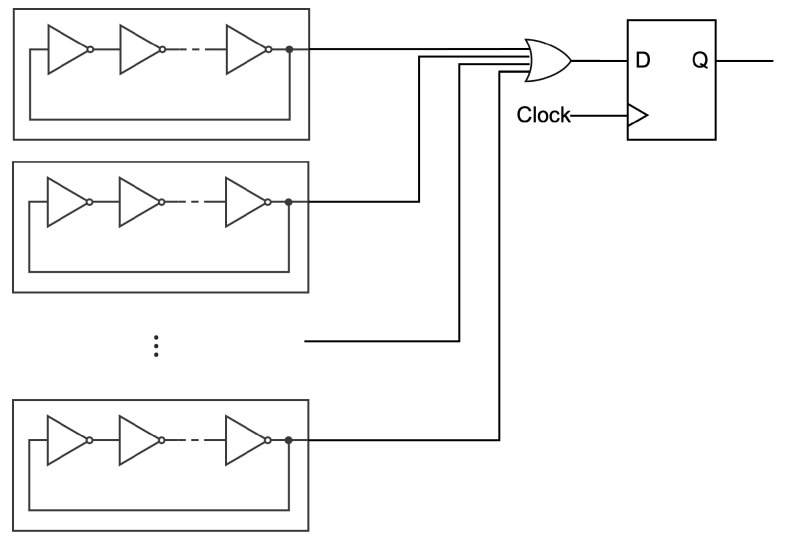
Illustration of the generator proposed by Sunar et al.

**Figure 4 entropy-24-00780-f004:**
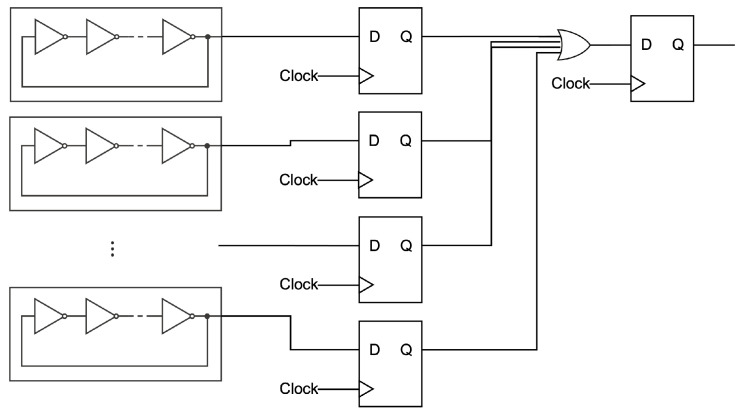
Illustration of the generator proposed by Wold et al.

**Figure 5 entropy-24-00780-f005:**
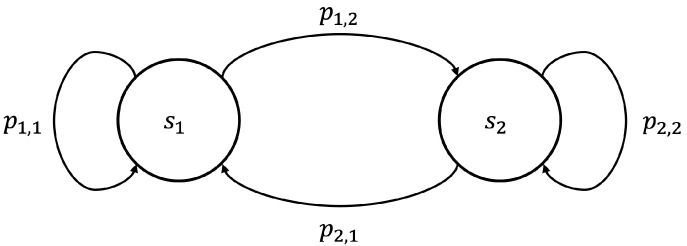
Example of a process diagram.

**Figure 6 entropy-24-00780-f006:**
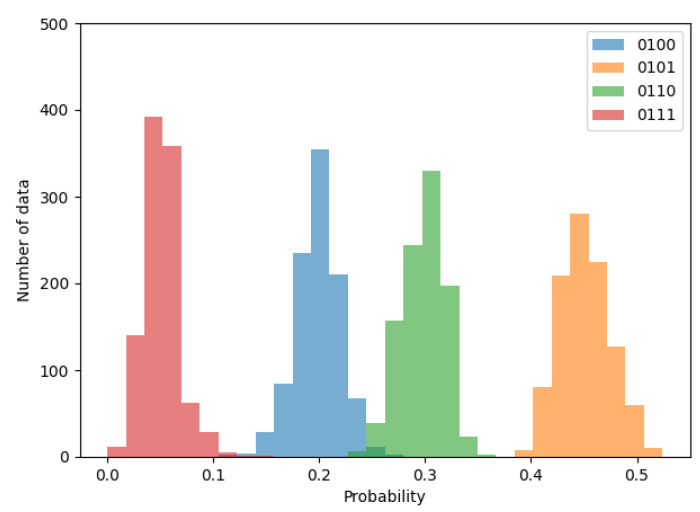
Histogram of transition probabilities when the number of ROs is 2.

**Figure 7 entropy-24-00780-f007:**
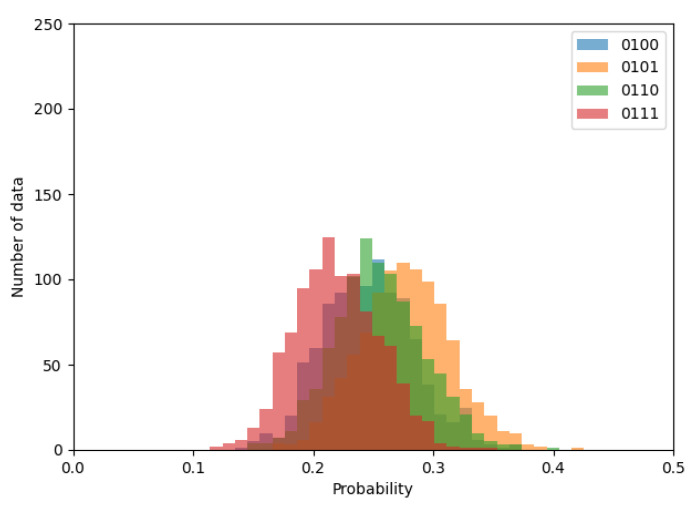
Histogram of transition probabilities when the number of ROs is 10.

**Figure 8 entropy-24-00780-f008:**
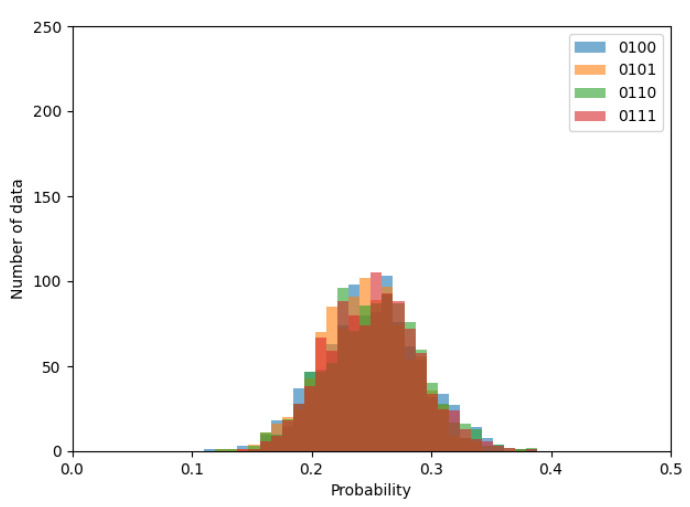
Histogram of transition probabilities when the number of ROs is 20.

**Figure 9 entropy-24-00780-f009:**
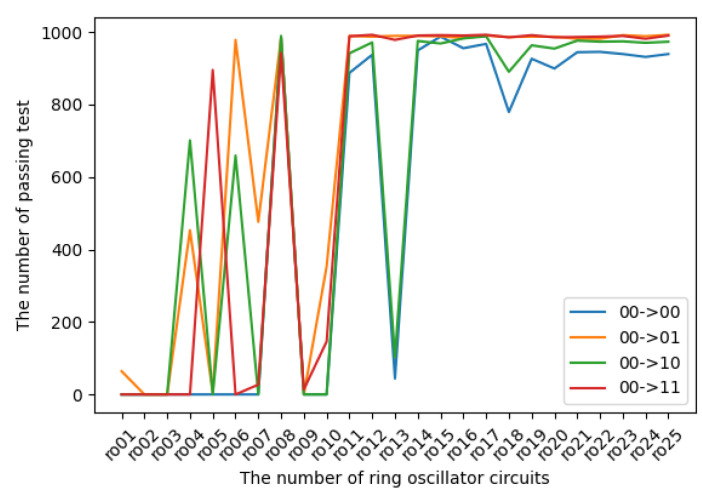
The transition probabilities from 00 (three NOT gates).

**Figure 10 entropy-24-00780-f010:**
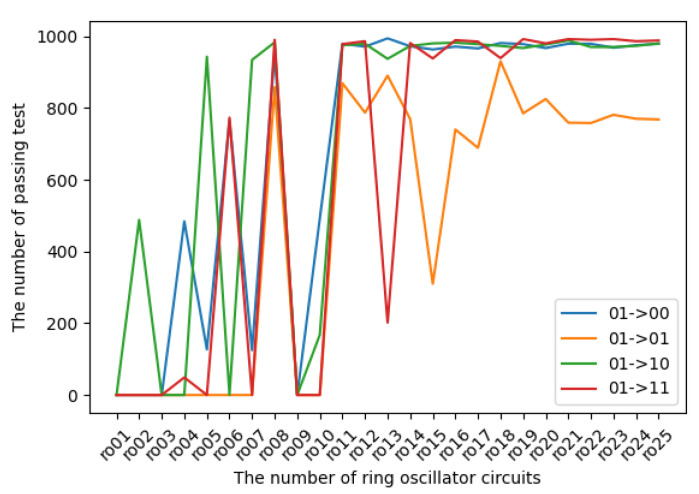
The transition probabilities from 01 (three NOT gates).

**Figure 11 entropy-24-00780-f011:**
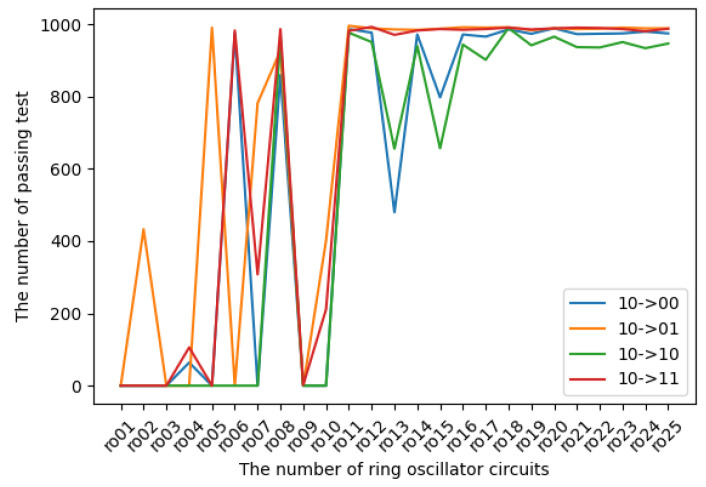
The transition probabilities from 10 (three NOT gates).

**Figure 12 entropy-24-00780-f012:**
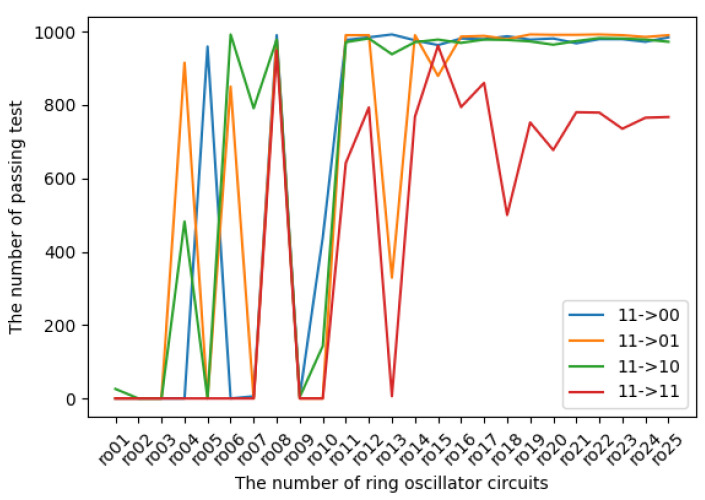
The transition probabilities from 11 (three NOT gates).

**Figure 13 entropy-24-00780-f013:**
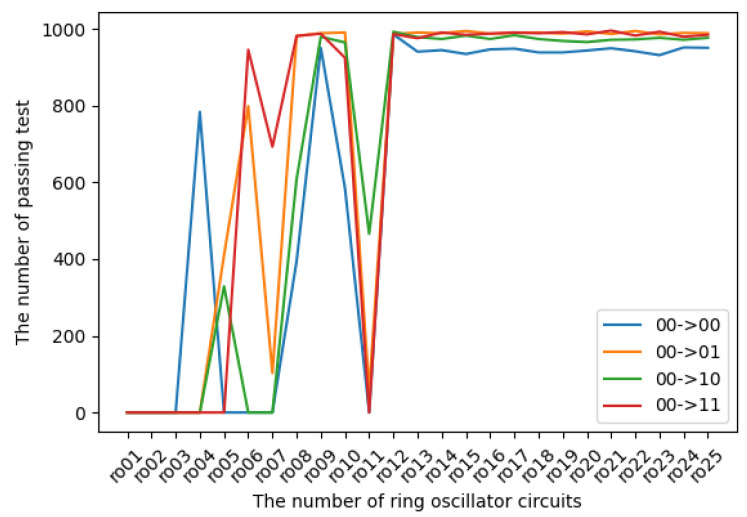
The transition probabilities from 00 (seven NOT gates).

**Figure 14 entropy-24-00780-f014:**
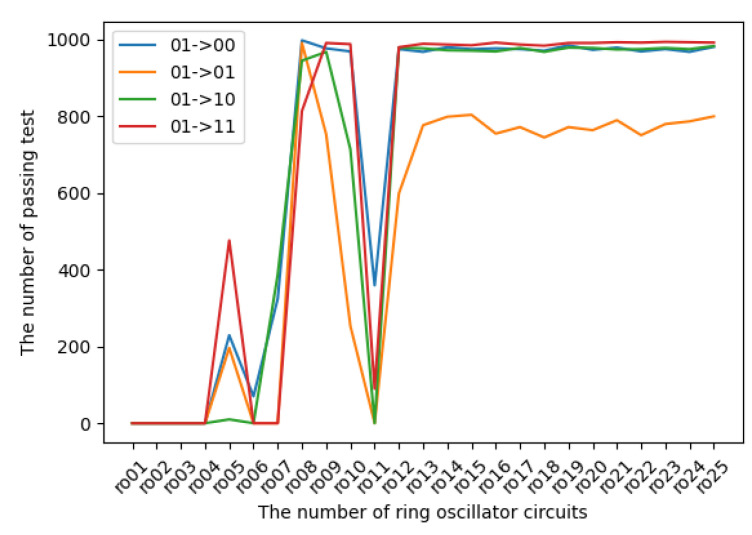
The transition probabilities from 01 (seven NOT gates).

**Figure 15 entropy-24-00780-f015:**
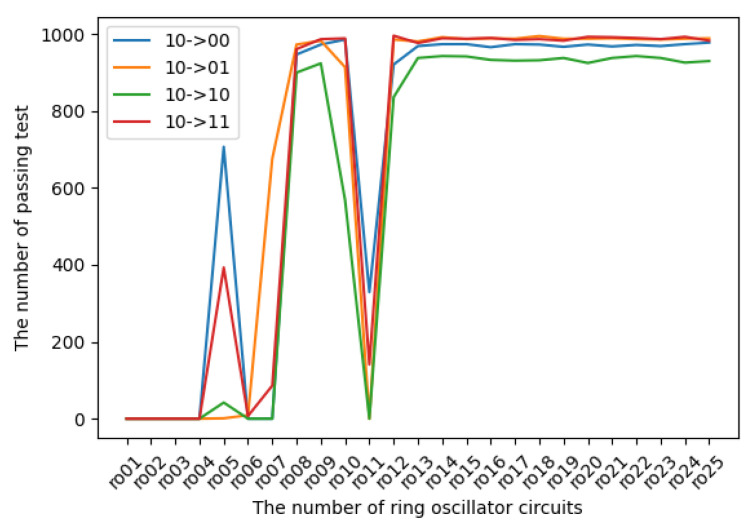
The transition probabilities from 10 (seven NOT gates).

**Figure 16 entropy-24-00780-f016:**
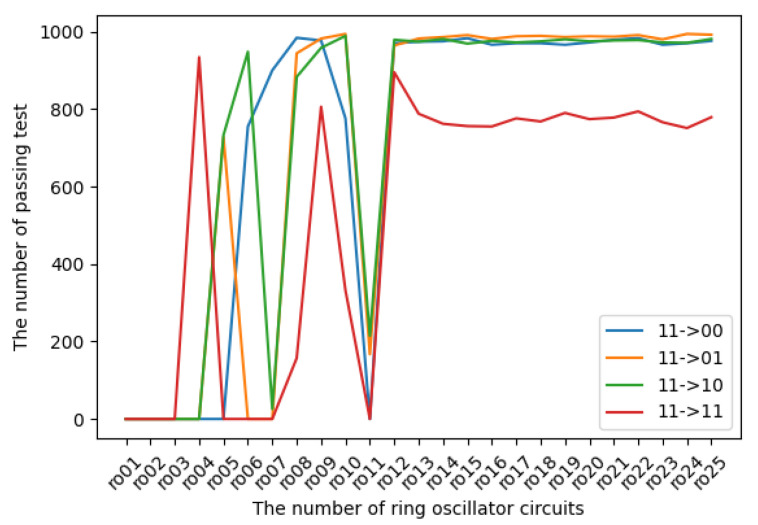
The transition probabilities from 11 (seven NOT gates).

**Table 1 entropy-24-00780-t001:** Comparisons of *p*-values.

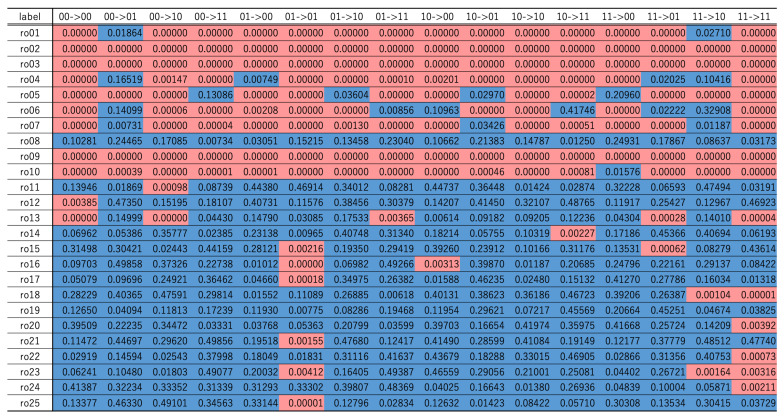

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
