# Peer review of "Transition Probability Test for an RO-Based Generator and the Relevance between the Randomness and the Number of ROs"

_entropy, 2022, doi:10.3390/e24060780_

Round 1
Reviewer 1 Report
Report
Title: Transition Probability Test for an RO-based Generator and a Relevance between the Randomness and the number of Ros
The authors focus on a Wold’s RO-based generator and propose a statistical 1 test. The test adopts the state transition probabilities in a Markov process and is designed to check 2 the uniformity of the probabilities based on a hypothesis testing. The reported results and analysis in this study are genuine, and they would be of great help in the present literature. However, I have the following observations:
- The authors should check the whole paper for possible typos and grammatical errors.
- Important outcomes of the present study must be highlighted in the abstract.
- What conclusions could be drawn from the results and analysis in this study?
- Write some future work for the conclusion section.
- Authors need to update the introduction section with some recent results in this field.
I strongly recommend the publication of this paper after addressing the above points by the authors.
Author Response
Thank you for your constructive comments.
The authors revised the points that the reviewer pointed out.
They are highlighted by red color text and the revised manuscript has been taken to the English proof service provided by MDPI.
Reviewer 2 Report
1- The authors must surly correct some grammatical and typographical errors that seen in the paper.
2-A Markov chain should be explained more
3-The authors shown the experimental results of the proposed method. How about the mathematical results?
Author Response
The authors appreciate the honorable reviewer’s comments.
We updated the explanation about the Markov chain.
Though the concrete mathematical result could not be obtained by this time result immediately, the authors believe the results would be helpful for clarifying the characteristic equation.
We mention it in the manuscript and the exploration is one of future work.
The manuscript has been taken to the English proof service provided by MDPI.
Reviewer 3 Report
The problem described in the paper seems to be new and worth dissemination. In general, my impression is quite good. The presentation quality is sufficient, so the paper could be considered for possible publication. I recommend a detailed proofreading of the whole manuscript to avoid linguistic errors and typos. In the paper, Z-test is proposed in which the standard deviation is known. However, in practice the standard deviation of the population (not of the sample) is not known and should be estimated. But in this case Z-test is not a correct choice: t-test must be used in which a t-Student test statistic is used. Please, refer this problem in the revised version.
Author Response
Thank you for your valuable comments.
The authors added an explanation of the problem and the revision is highlighted by red-colored texts.
Also, the manuscript has been taken to the English proof service provided by MDPI.